# ENHANCED GRADIENT ALIGNED CONTINUAL LEARNING VIA PARETO OPTIMIZATION

## ABSTRACT

Catastrophic forgetting remains a core challenge in continual learning (CL), whereby the models struggle to retain previous knowledge when learning new tasks. While existing gradient-alignment-based CL methods have been proposed to tackle this challenge by aligning gradients between previous and current tasks, they do not carefully consider the interdependence between previously learned tasks and fully explore the potential of seen tasks. Against this issue, we first adopt the MiniMax theorem and reformulate the existing commonly-adopted gradient alignment optimization problem in a gradient weighting framework. Then we incorporate the Pareto optimality to capture the interrelationship among previously learned tasks, and design a Pareto regularized gradient alignment algorithm (PRGA), which effectively enhances the overall performance of past tasks while ensuring the performance of the current task. Comprehensive empirical results demonstrate that the proposed PRGA outperforms current state-of-the-art continual learning methods across multiple datasets and different settings.

## 1 INTRODUCTION

An ideal intelligent system should possess the ability to incrementally learn, swiftly adapting to environmental changes while retaining previously acquired knowledge. Despite the remarkable performance of current deep neural networks (DNNs) on specific tasks, they still encounter challenges when it comes to effectively adapting to streaming tasks. One critical issue is catastrophic forgetting, whereby acquiring knowledge on a new task leads to a significant decline in performance on previously learned tasks. To alleviate this issue, numerous algorithms have been proposed in the field of continual learning (CL), aiming to enhance the incremental learning ability of DNNs on streaming tasks (Lopez-Paz & Ranzato, 2017; Kirkpatrick et al., 2017; Serra et al., 2018; Gupta et al., 2020; Guo et al., 2020; Arani et al., 2021; Wang et al., 2023; Chrysakis & Moens, 2023).

Gradient alignment (GA) is currently a simple-yet-effective research line in continual learning. It primarily focuses on directly manipulating the gradient of the current task to discover a gradient update direction that improves its performance, while simultaneously ensuring that the performance of previously learned tasks is not negatively affected (Lopez-Paz & Ranzato, 2017; Chaudhry et al., 2018; Guo et al., 2020; Riemer et al., 2019; Gupta et al., 2020). For example, the representative method GEM (Lopez-Paz & Ranzato, 2017) utilizes a small memory buffer to store samples from previous tasks and aims to find a gradient update direction $u$ adhering to two primary two main constraints: 1) the to-be-estimated $u$ should be as close as possible to the gradient of the current task $g_n$ for improving the performance on the current task; 2) the inner product $\langle u, g_i \rangle$ between the to-be-updated gradient $u$ and the gradient of every past task $g_i$, where $1 \leq i < n$, should be non-negative to prevent adverse effect on the performance of past task, thereby alleviating forgetting. To accelerate the optimization process of GEM (Lopez-Paz & Ranzato, 2017), instead of computing the individual gradient of each previous task, AGEM (Chaudhry et al., 2018) and MEGA (Guo et al., 2020) have proposed to compute an average past gradient $g_{avg}$ to execute the aforementioned inner-product constraint. The average past gradient $g_{avg}$ is calculated using examples randomly sampled from the memory buffer. Similarly, MER (Riemer et al., 2019) and La-MAML (Gupta et al., 2020) attempt to align the gradient of the current task $g_n$ with the average gradient $g_{avg}$ via a bi-level optimization procedure. As seen, these existing gradient alignment methods primarily focus on aligning the to-be-estimated gradient update direction $u$ for the current task with the gradients of previous tasks, $e.g.$, $g_i$ or $g_{avg}$, for preventing the performance deterioration on previous tasks.

Despite the promising success achieved by most existing gradient-alignment-based CL methods, there remains potential for further performance improvement. One main limitation of these methods is that they do not fully explore the mutual influence among previously learned tasks. Recent studies (Sener & Koltun, 2018; Lin et al., 2019; Momma et al., 2022), have shown that even seemingly unrelated tasks can exhibit strong dependencies and that different tasks can be viewed as an inductive bias in the learning system. If we can effectively utilize the inter-task dependencies, we can improve the retention of prior knowledge, mitigate the deleterious effects of forgetting, and ultimately elevate the overall performance across all tasks. Motivated by this insight, without sacrificing the performance of the current task, we aim to construct a new gradient alignment framework that models the relationships among previously learned tasks and manipulates the gradient update direction $u$ to maximize the overall performance of past tasks during training, rather than just adhering to the non-negative inner product constraints.

To achieve this goal, in this paper, we first revisit the current prevailing gradient-alignment-based CL methods and theoretically reformulate the optimization objective involved in these methods as a gradient weighting problem. From this perspective, we propose a Pareto regularized gradient alignment framework, called PRGA, which focuses on weighted updates of the gradients of past tasks to maximize their overall performance. The proposed PRGA not only enables effective learning of the current task but also captures the relationship among different previous tasks and balances the overall performance of past tasks. In summary, our main contributions are listed as follows:

- **New Perspective.** We mathematically derive that the objective of the existing gradient alignment pipeline is equivalent to a gradient weighting framework. Furthermore, we establish that current representative gradient-alignment-based CL methods, including GEM (Lopez-Paz & Ranzato, 2017), AGEM (Chaudhry et al., 2018), MEGA (Guo et al., 2020), and La-MAML (Gupta et al., 2020) can all be interpreted as special cases of the proposed gradient weighting framework.

- **Effective Algorithm.** Based on the derived gradient weighting framework, we propose to introduce the Pareto optimality mechanism to optimize the weights imposed on the gradient of every past task for maximizing the overall performance of all the previously learned tasks. Consequently, the gradient update $u$ not only considers the performance of the current task but also accounts for the interdependence among the previously learned tasks.

- **Superior Performance.** We conduct comprehensive experiments under different settings on three datasets, which validate the effectiveness of the proposed PRGA algorithm. Additionally, we conduct extensive ablation studies to investigate the effect of each component in PRGA algorithm.

## 2 RELATED WORKS

**Continual learning methods.** In the field of continual learning, various approaches have been proposed to address catastrophic forgetting in recent years. These existing approaches can be broadly categorized into four classes: regularization-based, parameter-isolation-based, replay-based, and gradient-alignment-based. Specifically, regularization-based methods (Kirkpatrick et al., 2017; Huszár, 2017; Ritter et al., 2018; Zenke et al., 2017; Yang et al., 2019; 2021) intend to design different regularization techniques to preserve important parameters for previously learned tasks. Parameter-isolation-based approaches (Serra et al., 2018; Mallya & Lazebnik, 2018; Fernando et al., 2017; Aljundi et al., 2017) focus on isolating task-specific parameters to prevent interference between tasks. Replay-based methods (Rolnick et al., 2019; Lopez-Paz & Ranzato, 2017; Chaudhry et al., 2018; Aljundi et al., 2019; Buzzega et al., 2020; Arani et al., 2021; Wang et al., 2023) try to maintain the knowledge acquired from previous tasks through different experience replay strategies, such as generating synthetic data or storing and replaying past experiences. Gradient-alignment-based approaches aim to align the gradients of previously learned tasks with that of the current task to alleviate catastrophic forgetting (Lopez-Paz & Ranzato, 2017; Chaudhry et al., 2018; Guo et al., 2020; Riemer et al., 2019; Gupta et al., 2020). In this paper, along the research line of gradient alignment, we propose a novel Pareto optimization based gradient-aligned framework for continual learning. Compared to the existing gradient alignment methods, our method additionally takes into account the interdependencies among past tasks and achieves an overall superior performance.

**Pareto Optimality.** As a crucial manner to achieve multi-objective optimization, Pareto optimality has been extensively investigated in multi-task learning (MTL) applications (Sener & Koltun, 2018; Lin et al., 2019; Fliege & Vaz, 2016; Mahapatra & Rajan, 2020), with the aim to balance different

competing tasks. To obtain Pareto optimality, MGDA (Sener & Koltun, 2018) proposed to convert the multi-objective optimization problem that accommodates all objectives of different tasks into a single-objective optimization problem. Afterwards, Lin et al. further explored Pareto fronts in MTL to ensure that the estimated solutions are uniformly distributed on the Pareto front. However, these Pareto optimization based MTL methods have not been fully investigated in the context of dynamic streaming tasks and are therefore unsuitable for CL tasks. In contrast, our proposed new gradient weighting framework is designed for the specific CL scenario, which makes it easy and natural to integrate the Pareto optimization mechanism for overall performance improvement. To the best of our knowledge, we are the first to introduce Pareto optimization specifically for CL.

## 3 PARETO REGULARIZED GRADIENT ALIGNMENT FOR CL

In this section, we will first revisit the current gradient alignment framework commonly adopted for continual learning and propose a reformulation of it as a gradient weighting method. Subsequently, we propose a specific gradient alignment framework for continual learning from the perspective of gradient weighting. Compared to existing gradient alignment methods, our proposed algorithm further models the relationships of the previously learned tasks to maximize the overall performance of those tasks. The details of the proposed framework are presented below.

### 3.1 REVISIT GRADIENT ALIGNMENT IN CONTINUAL LEARNING

Suppose there are $N$ sequential tasks $\{\mathcal{T}_1, \mathcal{T}_2, ..., \mathcal{T}_N\}$. During the streaming learning process, to obtain the update on the current task $\mathcal{T}_n$ with gradient $\boldsymbol{g}_n$, most existing gradient-alignment-based CL methods focus on identifying the next gradient update direction $\boldsymbol{u}$ that is most proximate to $\boldsymbol{g}_n$ for ensuring the performance of the current task without negatively affecting all the learned $n-1$ tasks in the past. Mathematically, the corresponding optimization problem can be formulated as (Lopez-Paz & Ranzato, 2017; Chaudhry et al., 2018):

$$\max_{\boldsymbol{u}} \; -\frac{1}{2}\|\boldsymbol{g}_n - \boldsymbol{u}\|_2^2, \quad s.t. \; \langle \boldsymbol{u}, \boldsymbol{g}_i \rangle \geq 0, \; i = 1, 2, \ldots, n-1, \tag{1}$$

where $\boldsymbol{g}_i$ represents the gradient of the previously learned task $\mathcal{T}_i$. The non-negative constraint (also known as regularization) represents that expected gradient $\boldsymbol{u}$ is updated in the direction forming an acute angle with every $\boldsymbol{g}_i$ in order to avoid deteriorating the performance of previous tasks.

By adopting the Lagrange multiplier (Boyd & Vandenberghe, 2004), we can convert Eq. (1) into an unconstrained form as:

$$\max_{\boldsymbol{u}} \min_{\lambda_i \geq 0} \; -\frac{1}{2}\|\boldsymbol{g}_n - \boldsymbol{u}\|_2^2 + \sum_{i=1}^{n-1} \lambda_i \langle \boldsymbol{u}, \boldsymbol{g}_i \rangle. \tag{2}$$

where $\lambda_i$ is a non-negative penalty coefficient. Let $\mathbb{Q}^{n-1} = \{(\lambda_1, ..., \lambda_{n-1}) | \lambda_i \geq 0\}$ and $\boldsymbol{g}(\boldsymbol{\lambda}) = \sum_{i=1}^{n-1} \lambda_i \boldsymbol{g}_i$, where $\boldsymbol{\lambda} = [\lambda_1, \ldots, \lambda_{n-1}]$. Eq. (2) can be equivalently written as:

$$\max_{\boldsymbol{u}} \min_{\boldsymbol{\lambda} \in \mathbb{Q}^{n-1}} \; -\frac{1}{2}\|\boldsymbol{g}_n - \boldsymbol{u}\|_2^2 + \langle \boldsymbol{u}, \boldsymbol{g}(\boldsymbol{\lambda}) \rangle. \tag{3}$$

From Eq. (3), we can know that the feasible domains of the maximum and the minimum optimization processes are both convex sets, and the objective function is convex w.r.t. the variable of the minimum operation for $\boldsymbol{\lambda}$ and concave w.r.t. the variable of the maximum operation for $\boldsymbol{u}$. According to the MiniMax theorem (see Appendix A.1), we can swap the order of the maximum and the minimum operation in Eq. (3) and then derive the following optimization problem as:

$$\min_{\boldsymbol{\lambda} \in \mathbb{Q}^{n-1}} \max_{\boldsymbol{u}} \; -\frac{1}{2}\|\boldsymbol{g}_n - \boldsymbol{u}\|_2^2 + \langle \boldsymbol{u}, \boldsymbol{g}(\boldsymbol{\lambda}) \rangle \quad \overset{(a)}{\Leftrightarrow} \min_{\boldsymbol{\lambda} \in \mathbb{Q}^{n-1}} \frac{1}{2}\|\boldsymbol{g}(\boldsymbol{\lambda})\|_2^2 + \langle \boldsymbol{g}_n, \boldsymbol{g}(\boldsymbol{\lambda}) \rangle$$

$$\overset{(b)}{\Leftrightarrow} \min_{\boldsymbol{\lambda} \in \mathbb{Q}^{n-1}} \frac{1}{2}\|\boldsymbol{g}(\boldsymbol{\lambda})\|_2^2 + \langle \boldsymbol{g}_n, \boldsymbol{g}(\boldsymbol{\lambda}) \rangle + \frac{1}{2}\|\boldsymbol{g}_n\|_2^2 \; \Leftrightarrow \min_{\boldsymbol{\lambda} \in \mathbb{Q}^{n-1}} \frac{1}{2}\|\boldsymbol{g}(\boldsymbol{\lambda}) + \boldsymbol{g}_n\|_2^2, \tag{4}$$

where (a) holds since the solution $\boldsymbol{u}^*$ of the maximum problem can be directly obtained as $\boldsymbol{u}^* = \boldsymbol{g}_n + \boldsymbol{g}(\boldsymbol{\lambda})$, and (b) holds since the term $\frac{1}{2}\|\boldsymbol{g}_n\|_2^2$ is independent of the optimization variable $\boldsymbol{\lambda}$.

As seen, the last step in Eq. (4) is a gradient weighting problem, which aims to learn the weight $\lambda_i$ imposed on the gradient of every past task $\mathcal{T}_i$. The solution $\boldsymbol{\lambda}^*$ can easily be solved by the Frank-Wolfe algorithm (Jaggi, 2013; Sener & Koltun, 2018) and then we can get the final update direction

as $\boldsymbol{u}^* = \boldsymbol{g}_n + \boldsymbol{g}(\boldsymbol{\lambda}^*)$. For clarity, we call the reformulated regularized gradient alignment method as RGA. Please refer to Appendix A.1 for the detailed optimization process.

**Remark 1:** Compared to the existing gradient aligned CL approaches, the proposed RGA framework has specific merits and contributions: 1) Based on the theoretical derivations (2)(3)(4), we carefully reformulate Eq. (1) as a gradient weighting optimization problem, making it possible to be easily solved with a higher computation efficiency than GEM (Lopez-Paz & Ranzato, 2017) which formulated (1) as a quadratic programming problem; 2) RGA encompasses AGEM (Chaudhry et al., 2018) as a special case where all previous tasks are merged into one task, *i.e.*, $n = 2$. Other gradient-alignment-based CL methods such as La-MAML (Gupta et al., 2020) and MEGA (Guo et al., 2020) can also be analyzed within our gradient weighting framework. In La-MAML, the involved optimization objective is equivalent to that of AGEM. In MEGA, the weights are manually assigned for different tasks. Please see Appendix A.2 for the details regarding the reformulation framework (4).

## 3.2 PARETO OPTIMIZATION-BASED GRADIENT ALIGNMENT FOR CONTINUAL LEARNING

As analyzed in Sec. 3.1, in RGA, the original gradient alignment optimization problem (1) is equivalent to the gradient weighting problem (4). By deeply exploring the derived optimal update direction $\boldsymbol{u}^* = \boldsymbol{g}_n + \boldsymbol{g}(\boldsymbol{\lambda}^*) = \boldsymbol{g}_n + \sum_{i=1}^{n-1} \lambda_i^* \boldsymbol{g}_i$, we can observe that: 1) the weighting coefficient on the gradient of the current task $\boldsymbol{g}_n$ is fixed and the to-be-optimized variable is the weight $\lambda_i$ imposed on the gradient of every previous task $\boldsymbol{g}_i$; 2) From the constraint in Eq. (1), $\lambda_i$ is solved under the regularization that the final gradient update direction $\boldsymbol{u}$ does not negatively impact the performance of the previously learned task $\mathcal{T}_i$. It is known that at every streaming training step, such a weighting-based gradient update rule for $\boldsymbol{u}$ is designed to improve the performance of the current task $\mathcal{T}_n$, while treating previous tasks as regularization tasks. This inspires us to ask the following question: is it possible to further optimize $\lambda_i$ to improve the overall performance of the previous tasks by taking into account the intrinsic correlation among them, rather than just considering them as auxiliary tasks? This section focuses on answering this question.

Based on the analysis in (Lin et al., 2019; Momma et al., 2022), it is known that seemingly unrelated tasks can exhibit significant dependencies. By effectively leveraging the dependencies among diverse tasks, it becomes possible to enhance the overall performance across all previously learned tasks. Motivated by this insight, we aim to model the relationships of past tasks to help optimize $\lambda_i$ for the entire performance improvement of previously seen tasks. To this end, we introduce the Pareto optimality to further optimize the weighting schemes $\boldsymbol{g}(\boldsymbol{\lambda})$ of previous tasks in RGA. Here Pareto optimality refers to solutions where the whole performance is not dominated by any single task, and it seeks to maximize the overall marginal benefit across different tasks (Sener & Koltun, 2018). In such a manner, we hope to thoroughly explore and model the interrelationships among previous tasks, thereby improving the overall performance of all these tasks.

Specifically, by considering $\boldsymbol{g}(\boldsymbol{\lambda})$ as the overall to-be-estimated gradient direction of all the previously learned tasks, our goal is to guarantee that $\boldsymbol{g}(\boldsymbol{\lambda})$ would benefit all the tasks learned so far. According to Pareto optimality, this can be achieved by maximizing the minimum inner product of the previous gradient $\boldsymbol{g}_i$ and $\boldsymbol{g}(\boldsymbol{\lambda})$, $i \in \{1, \ldots, n-1\}$. Mathematically, this can be formulated as:

$$\max_{\boldsymbol{g}(\boldsymbol{\lambda})} \min_{1 \le i \le n-1} \langle \boldsymbol{g}_i, \boldsymbol{g}(\boldsymbol{\lambda}) \rangle - \frac{1}{2} \|\boldsymbol{g}(\boldsymbol{\lambda})\|_2^2. \tag{5}$$

where the second term in the objective function is to constrain the gradient norm for avoiding infinity.

Define $\boldsymbol{g}(\tilde{\boldsymbol{\lambda}}) = \sum_{i=1}^{n-1} \tilde{\lambda}_i \boldsymbol{g}_i$ and $\mathbb{P}^{n-1} = \{(\tilde{\lambda}_1, ..., \tilde{\lambda}_{n-1}) | \tilde{\lambda}_i \ge 0, \sum_{i=1}^{n-1} \tilde{\lambda}_i = 1\}$. Then for the first term in Eq. (5) we prove that

$$\min_{1 \le i \le n-1} \langle \boldsymbol{g}_i, \boldsymbol{g}(\boldsymbol{\lambda}) \rangle \Leftrightarrow \min_{\tilde{\boldsymbol{\lambda}} \in \mathbb{P}^{n-1}} \langle \boldsymbol{g}(\tilde{\boldsymbol{\lambda}}), \boldsymbol{g}(\boldsymbol{\lambda}) \rangle, \tag{6}$$

based on the following two inequalities: As $\tilde{\boldsymbol{\lambda}} \in \mathbb{P}^{n-1}$, it always holds that $\langle \boldsymbol{g}(\tilde{\boldsymbol{\lambda}}), \boldsymbol{g}(\boldsymbol{\lambda}) \rangle \ge \min_{1 \le i \le n-1} \langle \boldsymbol{g}_i, \boldsymbol{g}(\boldsymbol{\lambda}) \rangle$. So we have $\min_{\tilde{\boldsymbol{\lambda}} \in \mathbb{P}^{n-1}} \langle \boldsymbol{g}(\tilde{\boldsymbol{\lambda}}), \boldsymbol{g}(\boldsymbol{\lambda}) \rangle \ge \min_{1 \le i \le n-1} \langle \boldsymbol{g}_i, \boldsymbol{g}(\boldsymbol{\lambda}) \rangle$; 2) Since $\langle \boldsymbol{g}_i, \boldsymbol{g}(\boldsymbol{\lambda}) \rangle$ is a special case of $\langle \boldsymbol{g}(\tilde{\boldsymbol{\lambda}}), \boldsymbol{g}(\boldsymbol{\lambda}) \rangle$, we can deduce that $\min_{\tilde{\boldsymbol{\lambda}} \in \mathbb{P}^{n-1}} \langle \boldsymbol{g}(\tilde{\boldsymbol{\lambda}}), \boldsymbol{g}(\boldsymbol{\lambda}) \rangle \le \min_{1 \le i \le n-1} \langle \boldsymbol{g}_i, \boldsymbol{g}(\boldsymbol{\lambda}) \rangle$.

By substituting Eq. (6) into Eq. (5), we can obtain that

$$\max_{\boldsymbol{g}(\boldsymbol{\lambda})} \min_{\tilde{\boldsymbol{\lambda}} \in \mathbb{P}^{n-1}} \langle \boldsymbol{g}(\tilde{\boldsymbol{\lambda}}), \boldsymbol{g}(\boldsymbol{\lambda}) \rangle - \frac{1}{2} \|\boldsymbol{g}(\boldsymbol{\lambda})\|_2^2. \tag{7}$$

---

**Algorithm 1** Frank-Wolfe Algorithm for Solving Eq. (8)

---

**Input:** Initialization $\tilde{\boldsymbol{\lambda}} = [\frac{1}{n-1}, \ldots, \frac{1}{n-1}]$
**Output:** the coefficient vector $\tilde{\boldsymbol{\lambda}}$ of previous tasks
 1: Precompute $\boldsymbol{D} = \boldsymbol{G}^{\mathrm{T}} \boldsymbol{G}$, where $\boldsymbol{G} = [\boldsymbol{g}_1^{\mathrm{T}}, \ldots, \boldsymbol{g}_{n-1}^{\mathrm{T}}]$
 2: **repeat**
 3:     $\boldsymbol{\alpha} = \mathrm{argmin}_{\boldsymbol{\alpha} \in \{\boldsymbol{\alpha}^{\mathrm{T}} \mathbf{1} = 1, \boldsymbol{\alpha} \succeq \mathbf{0}\}} \boldsymbol{\alpha}^{\mathrm{T}} \boldsymbol{D} \tilde{\boldsymbol{\lambda}}$
 4:     $\eta = \mathrm{argmin}_{\eta \in [0,1]} \left(\tilde{\boldsymbol{\lambda}} + \eta \left(\boldsymbol{\alpha} - \tilde{\boldsymbol{\lambda}}\right)\right)^{\mathrm{T}} \boldsymbol{D} \left(\tilde{\boldsymbol{\lambda}} + \eta \left(\boldsymbol{\alpha} - \tilde{\boldsymbol{\lambda}}\right)\right)$
 5:     $\tilde{\boldsymbol{\lambda}} \leftarrow (1 - \eta) \tilde{\boldsymbol{\lambda}} + \eta \boldsymbol{\alpha}$
 6: **until** $\eta \sim 0$ **or** Reaching the maximum iteration number

---

According to the MiniMax theorem, we can swap the order of minimization and maximization operations in Eq. (7). Similar to the derivations in Eq. (4), we can get the solution of the maximum optimization problem as $\boldsymbol{g}(\boldsymbol{\lambda}^*) = \boldsymbol{g}(\tilde{\boldsymbol{\lambda}})$ and then the minimization problem can be expressed as:

$$\min_{\tilde{\boldsymbol{\lambda}} \in \mathbb{P}^{n-1}} \frac{1}{2} \|\boldsymbol{g}(\tilde{\boldsymbol{\lambda}})\|_2^2 \Leftrightarrow \min_{\tilde{\boldsymbol{\lambda}} \in \mathbb{P}^{n-1}} \frac{1}{2} \|\textstyle\sum_{i=1}^{n-1} \tilde{\lambda}_i \boldsymbol{g}_i\|_2^2, \tag{8}$$

which can be easily solved by utilizing the Frank-Wolfe algorithm (Jaggi, 2013; Sener & Koltun, 2018) as listed in Alg. 1. Then, the gradient update direction is $\boldsymbol{u}^* = \boldsymbol{g}_n + \boldsymbol{g}(\tilde{\boldsymbol{\lambda}}^*)$. We refer to this derived Pareto regularized gradient alignment algorithm for CL as PRGA.

The gradient update rule $\boldsymbol{u}^* = \boldsymbol{g}_n + \boldsymbol{g}(\tilde{\boldsymbol{\lambda}}^*)$ tells that the gradient of the current task $\boldsymbol{g}_n$ plays an important role in the update $\boldsymbol{u}^*$ since its weighting coefficient is fixed as 1 while the weight $\tilde{\lambda}_i^*$ on the gradient of every previous task $\boldsymbol{g}_i$ meets $0 \le \tilde{\lambda}_i^* \le 1$ and $\sum_{i=1}^{n-1} \tilde{\lambda}_i^* = 1$. This means that our PRGA is a CL framework that prioritize different tasks. It principally focuses on the performance of the current task and simultaneously balances the entire performance improvement of past tasks.

**Remark 2:** Compared to GEM and RGA, the proposed PRGA not only integrates the gradient alignment mechanism to avoid forgetting and ensure the performance of the current task, but also models the mutual influence among different previous tasks in order to further improve the entire performance of the previously learned tasks via the Pareto optimization. It should be worth mentioning that it is exactly the gradient weighting perspective proposed in our RGA that makes it possible to further design a flexible weighting algorithm PRGA for the whole performance improvement. From this point, the proposed RGA and PRGA both have specific contributions, which will be validated in Sec. 4. Moreover, we also provide convergence analysis in Appendix B.

### 3.3 GRADIENT COMPUTATION

As evidenced by the theorectical analysis and empircal results in (Riemer et al., 2019; Gupta et al., 2020), aligning adaptation-based hyper-gradients among tasks yields superior performance compared to aligning the vanilla gradient. Motivated by this observation, we propose to compute $\boldsymbol{g}_i$ via the hyper-gradient manner for our proposed RGA in Sec. 3.1 and PRGA in Sec. 3.2. Concretely, at the $t^{\text{th}}$ training step, the hyper-gradient $\boldsymbol{g}_i, i \in \{1, \ldots, n\}$ is computed as:

$$\boldsymbol{g}_i = \frac{\partial \mathcal{L}(f_{\tilde{\boldsymbol{\theta}}_{t+1}}(x^m), y^m)}{\partial \boldsymbol{\theta}_t}, \quad \text{where } \tilde{\boldsymbol{\theta}}_{t+1} = \boldsymbol{\theta}_t - \alpha \nabla_{\boldsymbol{\theta}_t} \mathcal{L}(x_i, y_i), \tag{9}$$

where $f(\cdot)$ denotes the model with parameter $\boldsymbol{\theta}$; $(x^m, y^m)$ represents the samples drawn from the buffer $\mathcal{M}$ which stores samples of seen tasks; $(x_i, y_i)$ are samples from the task $\mathcal{T}_i$; $\mathcal{L}(\cdot)$ is the loss function; $\alpha$ denotes the adaptation learning rate and $\tilde{\boldsymbol{\theta}}_{t+1}$ denotes the intermediate parameters computing $\boldsymbol{g}_i$. The overall implementation algorithm of the proposed PRGA is outlined in Alg. 2.

## 4 EXPERIMENTS

In this section, we conduct comprehensive experiments to evaluate the effectiveness of our proposed methods based on diverse benchmark datasets and different CL settings. Besides, we provide a series of ablation studies to analyze and evaluate the specific role of each component in our method.

---

**Algorithm 2** The Entire Algorithm Implementation for the Proposed PRGA

---

**Input:** At the $t^{\text{th}}$ streaming training step, current training task $\mathcal{T}_n$, memory buffer $\mathcal{M}$, learning rates $\alpha$ and $\beta$, network parameter $\boldsymbol{\theta}_t$ for classification

**Output:** $\boldsymbol{\theta}_{t+1}$

1: Sample from memory buffer: $(x^m, y^m) \sim \mathcal{M}$
2: Sample from memory buffer for previous task $\mathcal{T}_i$: $(x_i, y_i) \sim \mathcal{M}_i, i \in \{1, 2, ..., n-1\}, \mathcal{M}_i \in \mathcal{M}$
3: Sample for the current task: $(x_n, y_n) \sim \mathcal{T}_n$
4: Compute the hyper-gradient $\boldsymbol{g}_i, i \in \{1, \ldots, n\}$ based on Eq. (9)
5: Compute the Pareto optimal weights $\tilde{\boldsymbol{\lambda}}^*$ based on Alg. 1 and get $\boldsymbol{g}(\tilde{\boldsymbol{\lambda}}^*) = \sum_{i=1}^{n-1} \tilde{\lambda}_i^* \boldsymbol{g}_i$
6: Compute the gradient update direction: $\boldsymbol{u}^* = \boldsymbol{g}_n + \boldsymbol{g}(\tilde{\boldsymbol{\lambda}}^*)$
7: Update network parameter: $\boldsymbol{\theta}_{t+1} = \boldsymbol{\theta}_t - \beta \boldsymbol{u}^*$
8: Update the memory buffer $\mathcal{M}$ with $(x_n, y_n)$ following (Buzzega et al., 2020)

---

## 4.1 EVALUATION PROTOCOL

**Benchmark Datasets.** Following (Buzzega et al., 2020; Arani et al., 2021; Wang et al., 2023), we select three widely-used datasets with varying complexity for the subsequent CL experiments, *i.e.*, Split CIFAR-10, Split CIFAR-100, and Split TinyImageNet (Buzzega et al., 2020). Split CIFAR-10 and Split CIFAR-100 are derived from the CIFAR-10 and CIFAR-100 (Krizhevsky et al., 2009), respectively. For Split CIFAR-10, the number of tasks $N$ is 5 and each task contains 2 classes. For Split CIFAR-100, $N$ is 20 and each task is composed of 5 classes. Split Tiny-ImageNet is divided into 20 tasks, each containing 10 classes. More details about datasets are included in Appendix C.

**Implementation Details.** Consistent to (Chrysakis & Moens, 2023; Lopez-Paz & Ranzato, 2017), we utilize the widely-adopted Reduced ResNet-18 (He et al., 2016) as the network backbone architecture to implement the proposed methods RGA and PRGA. During the training, the stochastic gradient descent (SGD) optimizer is used for optimizing network parameters and the batch size is set as 32. For the experiments on different datasets and various CL settings, the learning rates $\alpha$ and $\beta$ are fixed as 0.03 and the sampling batch size for memory buffer $\mathcal{M}$ as 300. Please note that all the comparison experiments are executed under the online class incremental setting, where each task is trained for only one epoch and the task identity is not provided during inference.

**Baselines.** For comprehensive comparisons, different types of state-of-the-art continual learning approaches are adopted, including regularization-based EWC (Huszár, 2017); rehearsal-based methods, such as, ER (Rolnick et al., 2019), DER (Buzzega et al., 2020), DER++ (Buzzega et al., 2020), CLSER (Arani et al., 2021), and ER-ACE (Caccia et al., 2021); and representative gradient alignment based AGEM (Chaudhry et al., 2018), GEM (Lopez-Paz & Ranzato, 2017), MER (Riemer et al., 2019), and La-MAML (Gupta et al., 2020). These comparing methods are primarily implemented based on the hyperparameter settings described in (Buzzega et al., 2020) or according to the hyperparameters specified in their respective papers.

**Evaluation Metrics.** To fairly and comprehensively validate the effectiveness of our proposed methods, we adopt several representative evaluation metrics for quantitative comparisons, including Average Accuracy, Forgetting Measure (Lopez-Paz & Ranzato, 2017), and Anytime Average Accuracy (Caccia et al., 2021). The higher these indicators, the better the performance. Specifically,

- **Average Accuracy (Acc)**: It represents the average accuracy on all the previously seen tasks after completing the model training on $N$ tasks, computed as $\text{Acc} = \text{Acc}_N = \frac{1}{N} \sum_{i=1}^{N} a_{i,N}$, where $a_{i,j}$ denotes the accuracy of the task $\mathcal{T}_i$ after the training on the task $\mathcal{T}_j$ and $\mathcal{T}_N$ is the last task.

- **Forgetting Measure (FM)**: This metric reflects the degree of forgetting that occurs in a model during sequential training. Concretely, it computes the average decrease from the best accuracy to the final accuracy after training on $\mathcal{T}_N$ across all $N$ tasks. This is denoted as $\text{FM} = \frac{1}{N} \sum_{i=1}^{N} (a_{i,N} - a_i^*)$, where $a_i^*$ is the best accuracy of the task $\mathcal{T}_i$ achieved during training.

- **Anytime Average Accuracy (AAA)**: Different from Acc, this indicator quantifies the classification performance of the model throughout the entire learning process. Specifically, its definition is $\text{AAA} = \frac{1}{N} \sum_{j=1}^{N} \text{Acc}_j = \frac{1}{N} \sum_{j=1}^{N} \left( \frac{1}{j} \sum_{i=1}^{j} a_{i,j} \right)$.

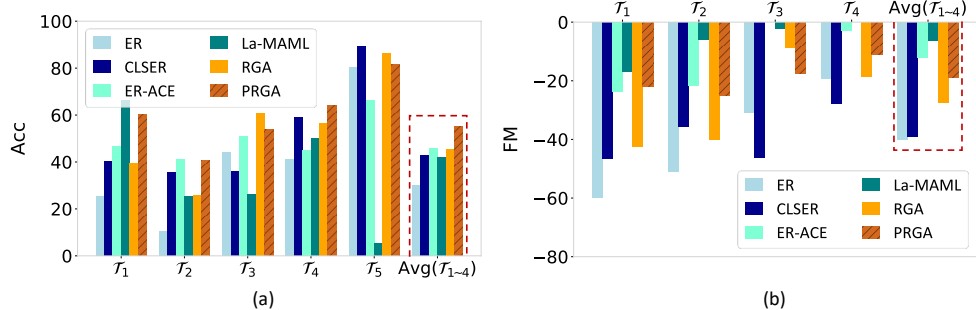

Figure 1: (a) accuracy (Acc) and (b) forgetting measure (FM) of each task achieved by different competitive CL methods on Split CIFAR-10 with $N = 5$ and buffer size $|\mathcal{M}|$ as 1k. Here $\mathcal{T}_{1\sim4}$ denotes the previously learned tasks and $\mathcal{T}_5$ is the current task, and Avg($\mathcal{T}_{1\sim4}$) means the average performance of past tasks $\mathcal{T}_{1\sim4}$.

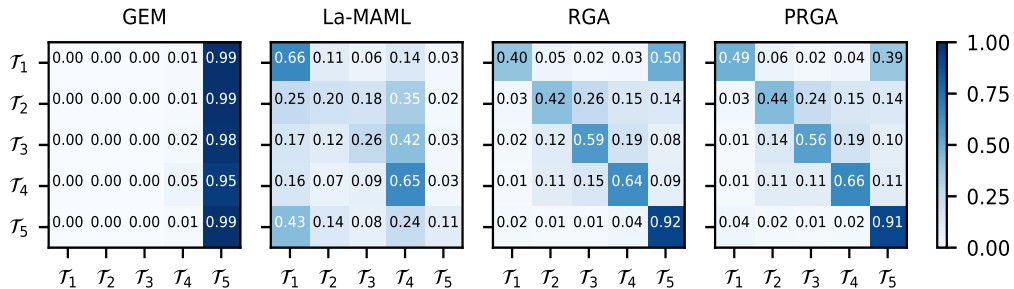

Figure 2: Task-based confusion matrix of various gradient-alignment-based CL methods on Split CIFAR-10 dataset with $|\mathcal{M}|$=1k.

## 4.2 EXPERIMENTAL RESULTS

**Direct verification about PRGA.** To better understand the working mechanism of our proposed PRGA, based on Split CIFAR-10, we first provide direct and intuitive model verification. Specifically, in Fig. 1(a), for the representative comparison methods, we provide the test accuracy of model on every task $\mathcal{T}_i$ ($i = 1, \ldots, 5$) after finishing the sequential training on all 5 tasks and the average accuracy on all the previously learned four tasks as Avg($\mathcal{T}_{1\sim4}$). We can find that: 1) Our proposed PRGA is obviously superior to other CL methods on Avg($\mathcal{T}_{1\sim4}$). This finely substantiates the potential of the proposed Pareto optimizatio-based gradient alignment to boost the entire performance of all the previously learned tasks, which complies with our design motivations; 2) On the newly learned $\mathcal{T}_5$, the proposed RGA and PRGA are comparable even superior to ER and CLSER. As seen, PRGA generally performs competitively on all the five tasks. Fig. 1(b) depicts the FM of different methods on every task. For Avg($\mathcal{T}_{1\sim4}$), PRGA averagely obtains a higher FM score and outperforms ER, CLSER, and RGA, which demonstrates the favorable capability in alleviating forgetting. Please note that for ER-ACE and La-MAML, they both over-emphasize preserving the performance of a past task without fully exploring the potential of the current task, which leads to a quite low $a_i^*$ and in turn an extremely high FM score, but a quite low accuracy on $\mathcal{T}_5$ as shown in Fig. 1(a). It is unfair to directly compare with these two methods. These results are consistent with (Caccia et al., 2021).

Moreover, we provide the task-based confusion matrix in Fig. 2 to investigate the proficiency of the proposed PRGA and other gradient alignment methods in managing task interrelationships. It is easily understood that a method that can balance task interrelationships and optimize the overall performance of all the tasks should, at a minimum, distinguish between tasks. By comparing the diagonal elements representing the correct classification probability of the task identity, we can find that in general, the proposed PRGA exhibits a larger value on all five tasks, and better balances every task, which comprehensively validates the effectiveness of the introduced Pareto component in capturing the interdependence among different tasks. More results are presented in Appendix D.

**Performance comparison with AAA and Acc.** Table 1 reports the average AAA and Acc of different CL methods on Split CIFAR-10, Split CIFAR-100, and Split-TinyImageNet, under different

Table 1: Performance comparison on benchmark datasets under different memory buffer sizes $|\mathcal{M}|$. All the results are averagely computed over 5 repetitions. '-' indicates the implementation is both highly time-consuming and unstable. The full table with 95% confidence interval is in Appendix D.

| Method | Split CIFAR-10 ($N=5$) | | | | Split CIFAR-100 ($N=20$) | | | | Split TinyImageNet ($N=20$) | | | |
|---|---|---|---|---|---|---|---|---|---|---|---|---|
| | $|\mathcal{M}| = 0.6$k | | $|\mathcal{M}| = 1$k | | $|\mathcal{M}| = 1$k | | $|\mathcal{M}| = 5$k | | $|\mathcal{M}|= 2$k | | $|\mathcal{M}| = 5$k | |
| | AAA | Acc | AAA | Acc | AAA | Acc | AAA | Acc | AAA | Acc | AAA | Acc |
| SGD | 34.04 | 16.68 | 34.04 | 16.68 | 9.67 | 3.24 | 9.67 | 3.24 | 7.63 | 2.17 | 7.63 | 2.17 |
| On-EWC | 36.51 | 18.37 | 36.51 | 18.37 | 9.87 | 2.77 | 9.87 | 2.77 | 7.96 | 2.43 | 7.96 | 2.43 |
| ER | 54.68 | 39.43 | 54.91 | 42.04 | 17.86 | 11.89 | 20.67 | 14.87 | 16.27 | 10.52 | 16.10 | 11.89 |
| DER | 49.06 | 25.80 | 48.25 | 23.90 | 10.96 | 3.71 | 10.47 | 3.68 | 8.04 | 2.46 | 7.65 | 2.04 |
| DER++ | 57.17 | 47.03 | 61.01 | 50.31 | 17.08 | 8.72 | 17.08 | 8.98 | 12.42 | 5.57 | 11.93 | 5.26 |
| CLSER | 61.64 | 50.36 | 63.27 | 53.06 | 22.58 | 15.68 | 23.25 | 16.42 | 18.50 | 10.03 | 18.88 | 11.61 |
| ER-ACE | 52.27 | 46.04 | 57.42 | 51.17 | 24.02 | 15.46 | 24.93 | 20.58 | 20.57 | **13.23** | 21.16 | 17.22 |
| AGEM | 37.67 | 18.51 | 37.62 | 18.03 | 10.61 | 3.75 | 10.80 | 3.52 | 7.66 | 2.33 | 7.79 | 2.40 |
| GEM | 37.78 | 18.84 | 37.00 | 18.73 | 13.43 | 6.04 | 13.71 | 6.46 | 10.17 | 3.70 | 10.27 | 3.81 |
| MER | 45.39 | 24.42 | 50.99 | 36.15 | – | – | – | – | – | – | – | – |
| La-MAML | 47.89 | 30.53 | 46.08 | 35.89 | 17.37 | 10.03 | 19.05 | 12.57 | 15.81 | 8.02 | 16.32 | 9.24 |
| RGA | 60.94 | 48.66 | 63.95 | 54.57 | 25.81 | 14.79 | 35.89 | 30.47 | 20.70 | 11.89 | 25.01 | 18.58 |
| PRGA | **63.62** | **53.42** | **66.23** | **58.50** | **26.68** | **16.54** | **36.34** | **33.36** | **21.56** | 12.69 | **25.48** | **19.40** |

Table 2: The Forgetting Measure (FM) with 95% confidence interval on benchmark datasets under different memory buffer size $|\mathcal{M}|$. All the reported results are averagely computed over 5 repetitions.

| Method | Split CIFAR-10 ($N=5$) | | Split CIFAR-100 ($N=20$) | | Split TinyImageNet ($N=20$) | |
|---|---|---|---|---|---|---|
| | $|\mathcal{M}| = 0.6$k | $|\mathcal{M}| = 1$k | $|\mathcal{M}| = 1$k | $|\mathcal{M}| = 5$k | $|\mathcal{M}| = 2$k | $|\mathcal{M}| = 5$k |
| SGD | $-61.01_{\pm 3.30}$ | $-61.01_{\pm 3.30}$ | $-54.24_{\pm 1.20}$ | $-54.24_{\pm 1.20}$ | $-43.58_{\pm 0.58}$ | $-43.58_{\pm 0.58}$ |
| On-EWC | $-64.21_{\pm 0.86}$ | $-64.21_{\pm 0.86}$ | $-56.55_{\pm 1.30}$ | $-56.55_{\pm 1.30}$ | $-44.30_{\pm 1.70}$ | $-44.30_{\pm 1.70}$ |
| ER | $-35.68_{\pm 3.20}$ | $-32.16_{\pm 5.10}$ | $-46.25_{\pm 0.47}$ | $-48.54_{\pm 1.10}$ | $-37.90_{\pm 0.28}$ | $-37.43_{\pm 0.81}$ |
| DER | $-54.60_{\pm 2.80}$ | $-55.22_{\pm 1.70}$ | $-60.58_{\pm 0.38}$ | $-59.94_{\pm 1.40}$ | $-47.11_{\pm 0.65}$ | $-46.27_{\pm 0.85}$ |
| DER++ | $-24.00_{\pm 1.30}$ | $-28.01_{\pm 1.31}$ | $-58.61_{\pm 0.60}$ | $-61.23_{\pm 0.19}$ | $-47.13_{\pm 2.00}$ | $-47.65_{\pm 0.76}$ |
| CLSER | $-29.03_{\pm 3.70}$ | $-31.38_{\pm 1.70}$ | $-45.63_{\pm 0.62}$ | $-50.71_{\pm 0.86}$ | $-44.45_{\pm 0.32}$ | $-44.21_{\pm 0.14}$ |
| AGEM | $-66.17_{\pm 1.61}$ | $-66.61_{\pm 1.60}$ | $-60.51_{\pm 0.34}$ | $-60.89_{\pm 0.37}$ | $-45.61_{\pm 0.04}$ | $-45.51_{\pm 0.27}$ |
| GEM | $-58.09_{\pm 3.90}$ | $-57.64_{\pm 2.70}$ | $-45.83_{\pm 0.69}$ | $-50.61_{\pm 1.90}$ | $-42.30_{\pm 0.11}$ | $-42.71_{\pm 0.03}$ |
| MER | $-32.78_{\pm 0.81}$ | $-23.33_{\pm 8.06}$ | – | – | – | – |
| RGA | $-27.88_{\pm 0.43}$ | $-21.99_{\pm 3.40}$ | $\mathbf{-35.23_{\pm 4.42}}$ | $-18.84_{\pm 1.77}$ | $-33.29_{\pm 1.70}$ | $-21.22_{\pm 0.54}$ |
| PRGA | $\mathbf{-23.27_{\pm 2.50}}$ | $\mathbf{-16.85_{\pm 2.80}}$ | $-37.61_{\pm 0.63}$ | $\mathbf{-17.55_{\pm 0.36}}$ | $\mathbf{-31.63_{\pm 1.60}}$ | $\mathbf{-20.22_{\pm 1.20}}$ |

memory buffer size $|\mathcal{M}|$, where the lower part below the horizontal line represents gradient-aligned CL methods. As seen, with the increase of $|\mathcal{M}|$, the performance of previous tasks can be better maintained and then almost all the comparing methods present an upward trend. Besides, from Split CIFAR-10 to Split CIFAR-100 to Split-TinyImageNet, as the task difficulty becomes higher, basically all the approaches show a downward trend. However, our proposed PRGA always achieves higher AAA and Acc scores, which almost consistently outperform other baselines across all the three datasets. This indicates that PRGA not only performs well on the final trained model but also maintains a sustained advantage throughout the entire streaming training process, which is crucial in CL. Compared to RGA, PRGA achieves higher performance gains, which finely substantiates the role of the Pareto optimization based gradient alignment in boosting the entire performance. Due to limited space, the results about 95% confidence intervals are provided in Appendix D.

**Performance comparison with FM.** Table 2 compares the performance of different CL methods in mitigating forgetting and lists the average FM with 95% confidence interval. As $|\mathcal{M}|$ increases, the FMs of other comparative methods improve very little, or even decrease. However, our proposed RGA and PRGA always show a significant performance improvement on different datasets. The underlying reason is that the derived hyper-gradient weighting formulation makes RGA and PRGA able to fully exploit the memory buffer for more accurate gradient alignment. For PRGA, while the regularization of Pareto optimality aims to help improve the performance of entire previous tasks (as verified in Table 1), from another perspective, this actually also avoids forgetting to a certain extent, thus helping PRGA obtain higher FM scores than RGA. As analyzed in Fig. 1, to avoid confusion, we defer reporting the FM results and analysis for ER-ACE and La-MAML to Appendix D.

**Performance comparisons on more realistic settings.** To comprehensively evaluate the effectiveness of our proposed methods, based on Split CIFAR-10, we additionally execute the comparing

Table 3: Performance comparison on the imbalanced Split CIFAR-10 with $|\mathcal{M}|$=1k under two different types of imbalanced CL settings, *i.e.*, Normal and Reversed.

| | Normal | | Reversed | | | Normal | | Reversed | |
|---|---|---|---|---|---|---|---|---|---|
| Methods | AAA | Acc | AAA | Acc | Methods | AAA | Acc | AAA | Acc |
| SGD | $36.51_{\pm0.40}$ | $16.38_{\pm0.33}$ | $35.32_{\pm0.87}$ | $17.74_{\pm0.22}$ | On-EWC | $38.95_{\pm0.25}$ | $16.95_{\pm0.31}$ | $37.82_{\pm0.27}$ | $17.91_{\pm0.19}$ |
| ER | $52.28_{\pm0.65}$ | $32.59_{\pm1.86}$ | $46.78_{\pm3.01}$ | $27.89_{\pm2.46}$ | A-GEM | $38.33_{\pm0.25}$ | $17.48_{\pm0.52}$ | $36.54_{\pm0.39}$ | $17.52_{\pm0.32}$ |
| DER | $47.09_{\pm1.33}$ | $16.89_{\pm0.69}$ | $40.37_{\pm1.54}$ | $18.12_{\pm0.63}$ | GEM | $41.36_{\pm0.44}$ | $18.03_{\pm0.53}$ | $38.71_{\pm1.08}$ | $18.24_{\pm0.35}$ |
| DER++ | $61.97_{\pm0.63}$ | $44.04_{\pm2.06}$ | $58.52_{\pm0.87}$ | $39.43_{\pm3.29}$ | MER | $54.61_{\pm1.39}$ | $35.15_{\pm1.01}$ | $52.24_{\pm1.92}$ | $39.47_{\pm1.77}$ |
| CLSER | $61.87_{\pm0.41}$ | $48.04_{\pm0.72}$ | $55.32_{\pm1.57}$ | $42.38_{\pm2.97}$ | La-MAML | $36.17_{\pm1.25}$ | $28.99_{\pm0.78}$ | $31.79_{\pm2.03}$ | $31.68_{\pm1.42}$ |
| ER-ACE | $61.47_{\pm1.42}$ | $44.12_{\pm2.33}$ | $60.21_{\pm0.17}$ | $48.16_{\pm1.79}$ | PRGA | $\mathbf{66.87}_{\pm\mathbf{1.92}}$ | $\mathbf{54.82}_{\pm\mathbf{1.55}}$ | $\mathbf{64.45}_{\pm\mathbf{1.38}}$ | $\mathbf{58.79}_{\pm\mathbf{2.66}}$ |

Table 4: Ablation study on the proposed PRGA. Here FW and HD are the abbreviations for Frank-Wolfe algorithm used for solving Eq. (4) and hypergradient derived in Sec. 3.3, respectively.

| Methods | FW | HD | Pareto | Split CIFAR-10 ($|\mathcal{M}|$=1k) | | | Split CIFAR-100 ($|\mathcal{M}|$=5k) | | | Split TinyImageNet ($|\mathcal{M}|$=5k) | | |
|---|---|---|---|---|---|---|---|---|---|---|---|---|
| | | | | AAA | Acc | FM | AAA | Acc | FM | AAA | Acc | FM |
| RGA$_{FW}$ | ✓ | | | 36.80 | 19.45 | -55.54 | 16.20 | 9.05 | -54.91 | 10.89 | 4.55 | -43.74 |
| RGA | ✓ | ✓ | | 63.95 | 54.57 | -21.99 | 35.89 | 30.47 | -18.84 | 25.01 | 18.58 | -21.22 |
| PRGA | ✓ | ✓ | ✓ | 66.23 | 58.50 | -16.85 | 36.34 | 33.36 | -17.55 | 25.48 | 19.40 | -20.22 |

Table 5: Performance on Split CIFAR-10 with 95% confidence interval on the smaller 3-layer DNN with $|\mathcal{M}|$=1k. The results are averagely computed over 5 runs.

| | Split-CIFAR10 ($|\mathcal{M}|$ =1k) | | | | | | |
|---|---|---|---|---|---|---|---|
| Methods | AAA | Acc | FM | Methods | AAA | Acc | FM |
| SGD | $34.96_{\pm0.27}$ | $16.08_{\pm0.10}$ | $-61.63_{\pm0.06}$ | On-EWC | $35.12_{\pm0.12}$ | $16.38_{\pm0.08}$ | $-60.65_{\pm0.24}$ |
| ER | $56.82_{\pm0.83}$ | $39.05_{\pm1.80}$ | $-35.83_{\pm0.81}$ | A-GEM | $35.88_{\pm0.09}$ | $14.15_{\pm0.27}$ | $-61.90_{\pm0.22}$ |
| DER | $50.31_{\pm0.19}$ | $29.04_{\pm0.22}$ | $-49.38_{\pm0.19}$ | GEM | $46.86_{\pm0.68}$ | $28.02_{\pm0.55}$ | $-43.39_{\pm1.17}$ |
| DER++ | $56.99_{\pm0.08}$ | $42.30_{\pm0.18}$ | $-34.07_{\pm0.24}$ | MER | $58.18_{\pm0.34}$ | $35.14_{\pm0.84}$ | $-34.37_{\pm0.74}$ |
| CLSER | $60.25_{\pm0.12}$ | $43.82_{\pm0.25}$ | $-33.97_{\pm0.04}$ | PRGA | $\mathbf{61.09}_{\pm\mathbf{0.62}}$ | $\mathbf{46.14}_{\pm\mathbf{0.43}}$ | $\mathbf{-22.13}_{\pm\mathbf{0.42}}$ |

experiments on two more realistic CL settings, including Normal class imbalanced CL and Reversed class imbalanced CL. Specifically, for the Normal setting, the number of samples possessed by each streaming task is in a decreasing order, while Reversed takes an increasing order. More details are included in Appendix D. The corresponding comparison results are reported in Table 3. As seen, even under these more challenging scenarios, our proposed PRGA still shows superior performance.

**Ablation study on each component of PRGA.** To evaluate the role of each component of PRGA, we conduct an ablation study based on Split CIFAR-10 with buffer size $|\mathcal{M}|$=1k. Table 4 presents the performance of different variants of the proposed gradient weighting framework, including RGA$_{FW}$, RGA, and PRGA. Here RGA$_{FW}$ represents the degraded version of RGA which adopts the vanilla gradient to implement $g_i$ instead of the hypergradient computed in Sec. 3.3. The subscript FW denotes the Frank-Wolfe algorithm utilized for solving Eq. (4). From the results, we can see that 1) The introduction of hypergardient helps RGA obtain higher performance than RGA$_{FW}$; 2) Compared to RGA, the proposed Pareto optimality strategy further brings performance improvement for PRGA.

**More comparisons on different backbones.** To explore the versatility of our method across different backbones, based on Split CIFAR-10 with $|\mathcal{M}|$=1k, we also incorporate a smaller three-layer convolutional network as an additional backbone for experimental analysis. The results are given in Table 5. Compared to Table 1 and Table 2 with Reduced ResNet-18 as backbone, although all the comparison methods show a relative performance degradation under the smaller backbone, PRGA still surpasses other CL methods on all the evaluation metrics and shows a fine applicability.

## 5 CONCLUSION

In this paper, for the continual learning task, we have adopted the MiniMax theorem and rationally reformulated the existing widely-adopted gradient alignment optimization problem in a gradient weighting framework. Such a novel perspective enables us to analyze most existing gradient alignment methods as special cases. Building on this insight, we have further proposed a Pareto regularized gradient alignment (PRGA) algorithm which considers the interrelationships among previous tasks with the aim of enhancing their collective performance. Comprehensive experiments across various datasets and settings have finely substantiated the superiority of the proposed PRGA as well as its good applicability beyond the current state-of-the-art continual learning methods.

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

# A APPENDIX

## A.1 THE MINIMAX THEOREM AND FRANK WOLFE ALGORITHM

**MiniMax Theorem.** (Du & Pardalos, 1995). This theorem answers the question of in which situation the order of minimization and maximization operations can be swapped. Concretely, let $X \in \mathbb{R}^n$ and $Y \in \mathbb{R}^m$ be two compact convex sets. Suppose $f : X \times Y \to \mathbb{R}$ is a continuous function that is concave-convex, i.e., $f(x, \cdot) : Y \to \mathbb{R}$ is convex for fixed $x$ and $f(\cdot, y) : X \to \mathbb{R}$ is concave for the fixed $y$. Then we can get that

$$\max_{x \in X} \min_{y \in Y} f(x, y) = \min_{y \in Y} \max_{x \in X} f(x, y). \tag{10}$$

As discussed in Section 3.1, the optimization problem described by Eq. (3) satisfies the above requirements. Therefore, we can interchange its optimization order and obtain the gradient weighting optimization problem, as depicted in Eq. (4).

**Frank Wolfe Algorithm.** (Jaggi, 2013; Sener & Koltun, 2018) This algorithm is an iterative first-order optimization algorithm for constrained convex optimization. Therefore, it can be employed to address the optimization issue presented in Eq. (4). The complete optimization procedure is detailed in Alg. 3.

---

**Algorithm 3** Frank-Wolfe Algorithm for Solving Eq. (4)

---

**Input:** Initialization $\boldsymbol{\lambda} = [\frac{1}{n-1}, \ldots, \frac{1}{n-1}]$.
**Output:** the coefficient vector $\boldsymbol{\lambda}$ of previous tasks.
1: **repeat**
2:      $\alpha = \text{argmin}_{i \in \{1,2,\ldots,n-1\}} \langle \boldsymbol{g}_i, \boldsymbol{g}_n + \sum_{i=1}^{n-1} \boldsymbol{g}_i \rangle$
3:      $\eta = \text{argmin}_\eta \| \boldsymbol{g}_n + \sum_{i=1}^{n-1} \boldsymbol{g}_i - \lambda_\alpha \boldsymbol{g}_\alpha + \eta \boldsymbol{g}_\alpha \|^2$
4:      $\boldsymbol{\lambda} = \boldsymbol{\lambda} - \lambda_\alpha \boldsymbol{I}_\alpha + \eta \boldsymbol{I}_\alpha.$      ▷ $\boldsymbol{I}_\alpha$ is a unit vector with a value of 1 at the $a$-th position.
5: **until** $\eta \sim 0$ or Reaching the maximum iteration.

---

## A.2 ANALYSIS OF THE PROPOSED RGA

By utilizing the MiniMax theorem, we have transformed the gradient alignment method into a gradient weighting problem, as shown in Eq. (4). The resulting optimization direction, determined by our proposed RGA, can be expressed as $\boldsymbol{u}^* = \boldsymbol{g}_n + \boldsymbol{g}(\boldsymbol{\lambda})$, where $\boldsymbol{\lambda}$ is solved to minimize the objective function $\frac{1}{2} \| \boldsymbol{g}(\boldsymbol{\lambda}) + \boldsymbol{g}_n \|_2^2$. To gain a deeper understanding of the optimization direction $\boldsymbol{u}^*$, we conduct an analysis of simple cases (i.e., $n = 2, 3$) in two-dimensional scenarios. These simplified cases allow us to examine the behavior of RGA in a more straightforward manner, facilitating a clear comparison with other gradient alignment methods.

**Comparing with Other Gradient Alignment Methods.** As depicted in Fig. 3 where $n = 2$, there are two different situations. According to our derived optimization objective as shown in Eq. (4), our goal is to get $\lambda_1$ such that $\| \lambda_1 \boldsymbol{g}_1 + \boldsymbol{g}_2 \|_2^2$ as small as possible. Concretely, in the situation (a) where $\langle \boldsymbol{g}_2, \boldsymbol{g}_1 \rangle \geq 0$, there is no interference between the two directions, leading to the final optimization direction $\boldsymbol{u}^*$ being $\boldsymbol{u}^* = 0 \cdot \boldsymbol{g}_1 + \boldsymbol{g}_2 = \boldsymbol{g}_2$. In contrast, in situation (b), where $\langle \boldsymbol{g}_2, \boldsymbol{g}_1 \rangle < 0$, interference between the two directions is observed. In this case, the final optimization direction is $\boldsymbol{u}^* = \boldsymbol{g}_2 - \frac{\langle \boldsymbol{g}_2, \boldsymbol{g}_1 \rangle}{\| \boldsymbol{g}_1 \|^2} \boldsymbol{g}_1$. It is worth noting that the solutions are exactly the same as those obtained from AGEM, which implies that AGEM can be regarded as a special case within our gradient weighting framework. Additionally, La-MAML, a simplified version of MER, has derived an equivalent form of its bi-level optimization goal that coincides with optimizing the CL objective of AGEM. This equivalence demonstrates that La-MAML can also be regarded as a specific instance or a special case of RGA. Similarly, MEGA just manually assigns weights to gradients of different tasks. Therefore, all these gradient alignment methods can be analyzed within this gradient weighting framework.

**Solving the Optimization Problem of RGA.** There is a closed-form solution when $n=2$, as shown in Fig. 3. For more general situations where $n \geq 2$, we can solve them using the Frank-Wolfe algorithm outlined in Alg. 3. To provide a better understanding, we also present the solutions for

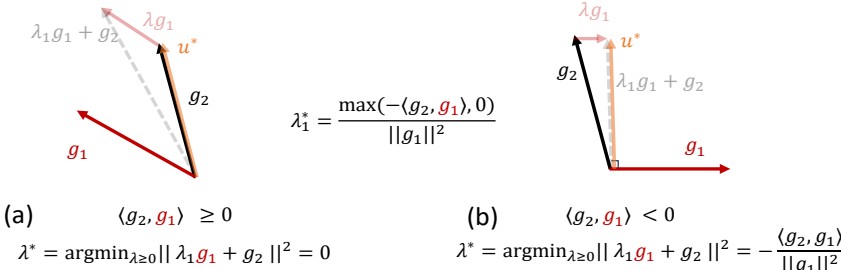

Figure 3: Illustration of the final determined optimization direction for our proposed RGA with two Tasks (i.e., n=2) in the simplest two-dimensional context.

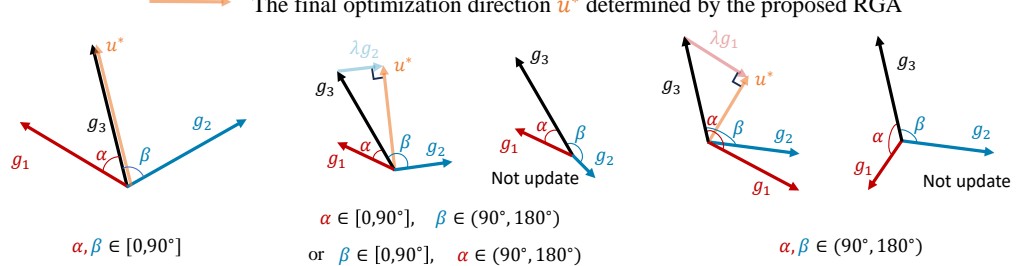

Figure 4: Illustration of the final optimization direction determined by our proposed RGA with three Tasks (i.e., $n=3$) in the simplest two-dimensional context.

the case when $n = 3$ in Fig. 4. It is important to note that when $\boldsymbol{g}_1, \boldsymbol{g}_2, \boldsymbol{g}_3$ are linearly dependent. This implies that no update occurs under such circumstances. Nevertheless, this linearly dependent situation is rare in high-dimensional gradient spaces, as the number of gradients is significantly smaller than the dimensions involved.

## B  CONVERGENCE ANALYSIS

To gain a deeper understanding of the proposed PRGA, we also provide its regret bound, as illustrated in Theorem 1. Let the notation $F_t(\theta) \triangleq \sum_{i=1}^{B_t} \mathcal{L}(f(x_i; \theta), y_i)$ and $F(\theta) = \sum_{t=1}^{T} F_t(\theta)$, where $B_t$ means the samples number of the $t$-th training batch, There are four requiring assumptions as follows:

**Assumption 1.** *The compact convex set $\mathcal{C} \subseteq \mathbb{R}^d$ has diameter $D$, i.e., $\forall \theta, \theta' \in \mathcal{C}, \|\theta - \theta'\| \le D$.*

**Assumption 2.** *Let $\boldsymbol{g}_t$ denote the $t$-th update gradient, there exisits $\sigma^2 < \infty$ such that $\|\nabla F_t(\theta) - \nabla F(\theta)\|^2 \le \sigma^2$.*

**Assumption 3.** *The difference of $F_t(\theta)$ and $F(\theta)$ is bounded over the constraint set $\mathcal{C}$, i.e., $\forall \theta \in \mathcal{C}, t \in \{1, ..., T\}$, there exists $M^2 < \infty$ such that with probability 1,*

$$\|F_t(\theta) - F(\theta)\|^2 \le M^2$$

**Assumption 4.** *The gradient $\nabla F_t(\theta)$ is unbiased and is L-Lipschitz continuous over the constraint set C, i.e.,*

$$\|\nabla F_t(\theta) - \nabla F_t(\theta')\| \le L\|\theta - \theta'\|, \forall \theta, \theta' \in \mathcal{C}$$

**Theorem 1.** *Let $T$ denote the training iteration. Suppose all the above four assumptions are satisfied, then with probability at least $1 - \delta$ for any $\delta \in (0, 1)$, we can get,*

$$R(T) \le (\log T + 1)(F(\theta_1) - F(\theta^*)) + \sigma^2 D \log T + \frac{LD^2(\log T + 1)^2}{2} + 4M\sqrt{T \log(8/\delta)}.$$

*Proof.* Then we will prove the regret bound $R(T)$ of PRGA as follows,

$$R(T) \triangleq \sum_{t=1}^{T} F_t(\theta_t) - \min_\theta \sum_{t=1}^{T} F_t(\theta). \tag{11}$$

Let $\theta^* = \mathrm{argmin}_\theta \sum_{t=1}^{T} F_t(\theta)$, then we can get,

$$
\begin{aligned}
R(T) &= \sum_{t=1}^{T} F_t(\theta_t) - \min_\theta \sum_{t=1}^{T} F_t(\theta) \\
&= \sum_{t=1}^{T} F_t(\theta_t) - \sum_{t=1}^{T} F_t(\theta^*) \\
&= \sum_{t=1}^{T} [F_t(\theta_t) - F_t(\theta^*)]
\end{aligned} \tag{12}
$$

Then, through calculating the term $F_t(\theta_t) - F_t(\theta^*)$, we can get the regret bound. We first define a sequence $s_t = \tilde{F}_t(\theta_t) - F_t(\theta^*) - (F(\theta_t) - F(\theta^*)), t = 1, ..., T$. It can be observed that $\mathbb{E}\left[s_t \mid \mathcal{F}_{t-1}\right] = 0$, where $\mathcal{F}_{t-1}$ is the $\sigma$-algebra generated by $\{F_1, \epsilon_1, ..., F_{t-1}, \epsilon_{t-1}\}$. It means that $\{s_t\}_{t=1}^{T}$ is a martingale difference sequence. According to Assumption 4, we have

$$\|s_t\| = \|F_t(\theta_t) - F_t(\theta^*) - (F(\theta_t) - F(\theta^*))\| \le 2M$$

Based on the Theorem 3.5 in Pinelis (1994), we can get

$$\mathbb{P}\left(\|\sum_{t=1}^{T} s_t\| \ge \lambda\right) \le 4\exp\left(-\frac{\lambda^2}{16TM^2}\right),$$

where $\lambda > 0$. By setting $\lambda = 4M\sqrt{T log(8/\delta)}$, we have with probability at least $1 - \delta/2$,

$$\sum_{t=1}^{T} s_t = \sum_{t=1}^{T}(F_t(\theta_t) - F_t(\theta^*)) - \sum_{t=1}^{T}(F(\theta_t) - F(\theta^*)) \le 4M\sqrt{T log(8/\delta)}.$$

Combining this term with Eq. (12), we can get,

$$R(T) \le \sum_{t=1}^{T}(F(\theta_t) - F(\theta^*)) + 4M\sqrt{T log(8/\delta)}$$

Let $\beta_t = \frac{1}{t}$ denote the learning rate, $\epsilon_t = g_t - \nabla F(\theta_t)$ then we can get,

$$
\begin{aligned}
F(\theta_t) - F(\theta^*) &= (1 - \beta_t)(F(\theta_{t-1}) - F(\theta^*)) + \alpha_t D\|\epsilon_{t-1}\|^2 + \frac{LD^2\beta_t^2}{2} \\
&\le \prod_{\tau=1}^{t-1}(1 - \beta_\tau)(F(\theta_1) - F(\theta^*)) + \sum_{\tau=1}^{t-1} \beta_\tau(D\|\epsilon_\tau\|^2 + \frac{LD^2\beta_\tau}{2}) \prod_{k=\tau+1}^{t-1}(1 - \beta_k) \\
&\le \frac{1}{t}(F(\theta_1) - F(\theta^*)) + \frac{D}{t}\sum_{\tau=1}^{t-1} \|\epsilon_\tau\|^2 + \frac{LD^2 \log t}{2t}
\end{aligned}
$$

$$
\begin{aligned}
\sum_{t=1}^{T} F(\theta_t) - F(\theta^*) &= \sum_{t=1}^{T} \frac{1}{t}(F(\theta_1) - F(\theta^*)) + \sum_{t=1}^{T} \frac{D}{t}\sum_{\tau=1}^{t-1} \|\epsilon_\tau\|^2 + \sum_{t=1}^{T} \frac{LD^2 \log t}{2t} \\
&\le (\log T + 1)(F(\theta_1) - F(\theta^*)) + \sigma^2 D \log T + \frac{LD^2(\log T + 1)^2}{2}
\end{aligned}
$$

Therefore, the final regret bound is,

$$R(T) \le (\log T + 1)(F(\theta_1) - F(\theta^*)) + \sigma^2 D \log T + \frac{LD^2(\log T + 1)^2}{2} + 4M\sqrt{T \log(8/\delta)}$$

$\square$

## C    More Details of the Experiments.

**Benchmark Datasets.** Both CIFAR-10 and CIFAR-100 datasets (Krizhevsky et al., 2009) consist of 50,000 training images and 10,000 testing images, all with a size of $32\times 32$ pixels. However, they differ in the number of classes: CIFAR-10 contains 10 classes, while CIFAR-100 includes 100 classes. Following (Chen et al., 2022), the CIFAR-10 and CIFAR-100 are evenly split into 5 tasks and 20 tasks respectively, denoted as Split CIFAR-10 and Split CIFAR-100. Similarly, the TinyImageNet dataset (Buzzega et al., 2020), which contains 200 classes and 100,000 images with a size of $64\times64$ pixels, is divided into 20 tasks, with each task containing 10 classes.

**Details of the Comparison Methods.** In this study, we compare the proposed PRGA method with other gradient alignment methods as well as other representative CL methods. Here is a brief introduction to these comparison methods:

*Gradient Alignment Methods*:

- **GEM** (Lopez-Paz & Ranzato, 2017). This method manipulates the gradient of the current task in a way that ensures the final update direction satisfies the non-negative inner-product constraints with the gradients of previous tasks.

- **AGEM** (Chaudhry et al., 2018). Due to GEM solves its constrained optimization problem through quadratic programming, which is time-consuming, AGEM opts to simplify the problem by merging all previous tasks into one. This enables the derivation of a closed-form solution.

- **MEGA** (Guo et al., 2020). This method proposes to manually assign weights to gradients of different tasks and then update the model in the weighted direction.

- **MER** (Riemer et al., 2019). This approach utilizes replay examples to align the gradients among previous tasks and the current task to maximize the transfer from previous tasks and minimize interference.

- **La-MAML** (Gupta et al., 2020). This method simplifies the optimization process of MER and shares a similar gradient alignment objective.

*Other Representative CL Methods*:

- **On-EWC** (Huszár, 2017). As one representative regularization-based CL method, it adopts a Fisher information matrix to approximate the Hessian to discern and protect the important weights.

- **ER** (Rolnick et al., 2019). It constructs a memory buffer to store samples of previous tasks and utilize them during the sequential training.

- **DER++** (Buzzega et al., 2020). Building upon the foundation of ER, DER++ stores the predicted logits of each example. This strategy is employed to apply distillation loss during the training process.

- **CLSER** (Arani et al., 2021). On the basis of ER, CLSER employs the dual memory which maintains short-term and long-term semantic memories.

- **ER-ACE** (Caccia et al., 2021). This method introduces a novel training approach that primarily focuses on previous tasks, aiming to avoid abrupt feature changes associated with these tasks.

## D    Other Experimental Results

### D.1    Full Table with Confidence Interval

In order to better present the results, we provide the mean performance of AAA, Acc, and FM metrics, along with their 95% confidence intervals, specifically for each dataset, as shown in Table 6, Table 7, and Table 8. The results clearly demonstrate that the proposed PRGA method consistently achieves the highest average performance in terms of AAA and Acc across all three datasets under different buffer sizes. With respect to the FM metric, aside from ER-ACE and La-MAML, methods that inherently have an extremely high $a_i^*$ as analyzed in Sec.4, our method also outperforms other methods, achieving the best FM performance.

Table 6: Performance of all the comparing methods on Split CIFAR-10 under different memory buffer size $|\mathcal{M}|$. All the reported results are averagely computed over 5 repetitions.

| Method | Split CIFAR-10 ($N = 5$) | | | | | |
| | $|\mathcal{M}| = 0.6k$ | | | $|\mathcal{M}| = 1k$ | | |
| | AAA | Acc | FM | Acc | AAA | FM |
|---|---|---|---|---|---|---|
| SGD | 34.04±3.10 | 16.68±0.27 | -61.01±3.30 | 34.04±3.10 | 16.68±0.27 | -61.01±3.30 |
| On-EWC | 36.51±0.75 | 18.37±0.30 | -64.21±0.86 | 36.51±0.75 | 18.37±0.30 | -64.21±0.86 |
| ER | 54.68±5.11 | 39.43±3.79 | -35.68±3.20 | 54.91±1.65 | 42.04±4.33 | -32.16±5.10 |
| DER | 49.06±1.30 | 25.80±0.51 | -54.60±2.80 | 48.25±1.95 | 23.90±0.47 | -55.22±1.70 |
| DER++ | 57.17±1.08 | 47.03±1.03 | -24.00±1.30 | 61.01±1.55 | 50.31±1.50 | -28.01±1.31 |
| CLSER | 61.64±1.62 | 50.36±4.06 | -29.03±3.70 | 63.27±0.71 | 53.06±1.58 | -31.38±1.70 |
| ER-ACE | 52.27±0.97 | 46.04±3.75 | -9.12±2.94 | 57.42±0.99 | 51.17±3.44 | -9.68±2.90 |
| AGEM | 37.67±1.76 | 18.51±0.07 | -66.17±1.61 | 37.62±1.53 | 18.03±0.43 | -66.61±1.60 |
| GEM | 37.78±1.98 | 18.84±1.17 | -58.09±3.90 | 37.00±0.47 | 18.73±0.76 | -57.64±2.70 |
| MER | 45.36±1.48 | 24.42±0.42 | -32.78±0.81 | 50.99±2.75 | 36.15±1.21 | -23.33±8.06 |
| La-MAML | 47.89±1.45 | 30.53±3.73 | -10.09±0.52 | 46.08±3.45 | 35.89±0.36 | -5.03±0.86 |
| RGA | 60.94±2.65 | 48.66±5.52 | -27.88±0.43 | 63.95±2.00 | 54.57±0.84 | -21.99±3.40 |
| PRGA | **63.62±2.65** | **53.42±1.17** | -23.27±2.50 | **66.23±1.73** | **58.50±2.43** | -16.85±2.80 |

Table 7: Performance of all the comparing methods on Split CIFAR-100 under different memory buffer size $|\mathcal{M}|$. All the reported results are averagely computed over 5 repetitions.

| Method | Split CIFAR-100 ($N = 20$) | | | | | |
| | $|\mathcal{M}| = 1k$ | | | $|\mathcal{M}| = 5k$ | | |
| | AAA | Acc | FM | Acc | AAA | FM |
|---|---|---|---|---|---|---|
| SGD | 9.67±0.18 | 3.24±0.12 | -54.24±1.20 | 9.67±0.18 | 3.24±0.12 | -54.24±1.20 |
| On-EWC | 9.87±0.28 | 2.77±0.27 | -56.55±1.30 | 9.87±0.28 | 2.77±0.27 | -56.55±1.30 |
| ER | 17.86±0.26 | 11.89±0.27 | -46.25±0.47 | 20.67±0.68 | 14.87±0.60 | -48.54±1.10 |
| DER | 10.96±0.20 | 3.71±0.11 | -60.58±0.38 | 10.47±0.32 | 3.68±0.06 | -59.94±1.40 |
| DER++ | 17.30±0.30 | 8.72±0.52 | -58.61±0.60 | 17.08±0.79 | 8.98±1.05 | -61.23±0.19 |
| CLSER | 22.58±0.42 | 15.68±0.62 | -45.63±0.62 | 23.25±1.34 | 16.42±1.52 | -50.71±0.86 |
| ER-ACE | 24.02±0.22 | 15.46±1.05 | -12.01±0.81 | 24.93±1.95 | 20.58±0.36 | -9.84±1.00 |
| AGEM | 10.61±0.09 | 3.75±0.18 | -60.51±0.34 | 10.80±0.15 | 3.52±0.13 | -60.89±0.37 |
| GEM | 13.43±0.04 | 6.04±0.52 | -45.83±0.69 | 13.71±0.23 | 6.46±0.93 | -50.61±1.90 |
| La-MAML | 17.37±0.59 | 10.03±0.25 | -5.95±0.01 | 19.05±0.54 | 12.57±0.34 | -6.45±0.33 |
| RGA | 25.81±0.51 | 14.79±0.71 | -35.23±4.42 | 35.89±0.84 | 30.47±0.43 | -18.84±1.77 |
| PRGA | **26.68±0.21** | **16.54±0.34** | -37.61±0.63 | **36.34±1.24** | **33.36±1.92** | -17.55±0.36 |

## D.2 OTHER ANALYSIS RESULTS

To better validate the effectiveness of the proposed PRGA, we also provide the accuracy of each task on Split CIFAR-100 and Split TinyImageNet, as shown in Fig. 6. It is evident from the results that our method, on average, covers a large area, thereby validating that our proposed PRGA can indeed sustain the performance of previous tasks and effectively mitigate forgetting. Moreover, we also provide the confusion matrix of other CL methods, including ER, DER++, CLSER, and ER-ACE. By comparing with the Fig. 2, it can be seen that all the four methods cannot better distinguish $\mathcal{T}_2$ and $\mathcal{T}_3$, while the proposed PRGA exhibits much better performance. This further confirms that our proposed PRGA, by introducing Pareto optimization to consider the interrelationships between past tasks, can effectively capture and model the relationships among the past tasks.

Table 8: Performance of all the comparing methods on Split TinyImageNet under different memory buffer size $|\mathcal{M}|$. All the reported results are averagely computed over 5 repetitions.

| Method | Split TinyImageNet ($N = 20$) | | | | | |
|---|---|---|---|---|---|---|
| | $|\mathcal{M}| = 2k$ | | | $|\mathcal{M}| = 5k$ | | |
| | AAA | Acc | FM | Acc | AAA | FM |
| SGD | $7.63 \pm 0.24$ | $2.17 \pm 0.07$ | $-43.58 \pm 0.58$ | $7.63 \pm 0.24$ | $2.17 \pm 0.07$ | $-43.58 \pm 0.58$ |
| On-EWC | $7.96 \pm 0.12$ | $2.43 \pm 0.13$ | $-44.30 \pm 1.70$ | $7.96 \pm 0.12$ | $2.43 \pm 0.13$ | $-44.30 \pm 1.70$ |
| ER | $16.27 \pm 0.10$ | $10.52 \pm 0.59$ | $-37.90 \pm 0.28$ | $16.10 \pm 0.24$ | $11.89 \pm 0.63$ | $-37.43 \pm 0.81$ |
| DER | $8.04 \pm 0.20$ | $2.46 \pm 0.06$ | $-47.11 \pm 0.65$ | $7.65 \pm 0.11$ | $2.04 \pm 0.15$ | $-46.27 \pm 0.85$ |
| DER++ | $12.42 \pm 0.34$ | $5.57 \pm 0.11$ | $-47.13 \pm 2.00$ | $11.93 \pm 0.34$ | $5.26 \pm 0.21$ | $-47.65 \pm 0.76$ |
| CLSER | $18.50 \pm 0.08$ | $10.03 \pm 0.22$ | $-44.45 \pm 0.32$ | $18.88 \pm 0.59$ | $11.61 \pm 0.19$ | $-44.21 \pm 0.14$ |
| ER-ACE | $20.57 \pm 0.49$ | $13.23 \pm 1.13$ | $-8.42 \pm 1.25$ | $21.16 \pm 0.14$ | $17.22 \pm 0.54$ | $-6.56 \pm 0.64$ |
| AGEM | $7.66 \pm 0.35$ | $2.33 \pm 0.18$ | $-45.61 \pm 0.04$ | $7.79 \pm 0.03$ | $2.40 \pm 0.22$ | $-45.51 \pm 0.27$ |
| GEM | $10.17 \pm 0.28$ | $3.70 \pm 0.44$ | $-42.30 \pm 0.11$ | $10.27 \pm 0.07$ | $3.81 \pm 0.15$ | $-42.71 \pm 0.03$ |
| La-MAML | $15.81 \pm 0.41$ | $8.02 \pm 0.10$ | $-3.76 \pm 0.11$ | $16.32 \pm 0.49$ | $9.24 \pm 0.34$ | $-4.38 \pm 0.50$ |
| RGA | $20.70 \pm 0.77$ | $11.89 \pm 0.54$ | $-33.29 \pm 1.70$ | $25.01 \pm 0.91$ | $18.58 \pm 0.04$ | $-21.22 \pm 0.54$ |
| PRGA | $\mathbf{21.56} \pm \mathbf{0.44}$ | $\mathbf{12.69} \pm \mathbf{0.47}$ | $-31.63 \pm 1.60$ | $\mathbf{25.48} \pm \mathbf{0.67}$ | $\mathbf{19.40} \pm \mathbf{0.92}$ | $-20.22 \pm 1.20$ |

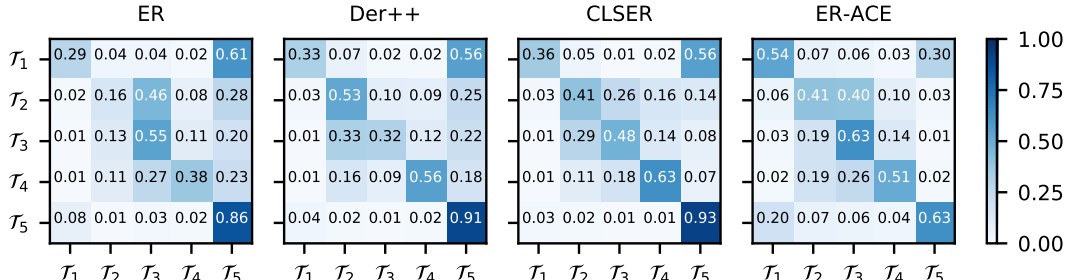

Figure 5: Task-based confusion matrix of various gradient-alignment-based CL methods on Split CIFAR-10 dataset with $|\mathcal{M}|$=1k.

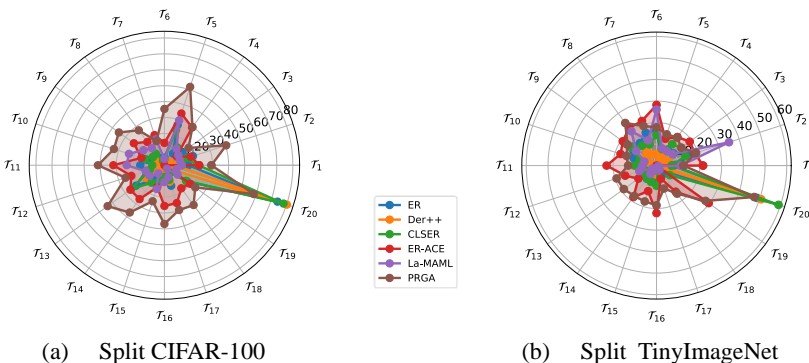

(a)  Split CIFAR-100          (b)  Split TinyImageNet

Figure 6: Radar charts illustrating each task accuracy on Split CIFAR-100 dataset and Split TinyImageNet with $|\mathcal{M}|$=5k.

