# OpenReview forum: "Enhanced Gradient Aligned Continual Learning via Pareto Optimization"
_ICLR.cc/2024/Conference — ICLR 2024 Conference Withdrawn Submission_

### Official Review · Reviewer_FMNS · 2023-10-15

**Soundness:** 3 good
**Presentation:** 3 good
**Contribution:** 2 fair
**Rating:** 5
**Confidence:** 5

**Summary:**

This paper begins by applying the MiniMax theorem and reformulating the commonly used gradient alignment optimization problem in continual learning within a gradient weighting framework. They find that existing continual learning methods ignore the interdependence among previously learned tasks. Subsequently, they integrate the concept of Pareto optimality to account for the connections among tasks that were previously learned. This leads to a Pareto regularized gradient alignment algorithm (PRGA), which effectively boosts the overall performance of tasks from the past, while maintaining the performance of the current task.  Extensive experiments on several datasets demonstrate the effectiveness of the proposed method.

**Strengths:**

* This paper is well-written and easy to follow.


* Introducing pareto optimality to continual learning seems reasonable.


* The performance improvement seem significant.

**Weaknesses:**

* The proposed methods  are very computationally costly. The main algorithm is a nested algorithm consists of two algorithms.  Furthermore, it is unclear about the complexity of  the Frank-Wolf algorithm applied to continual learning. It would be better to provide some complexity analysis and running time comparisons.


* The proposed method is similar to weighting the different tasks in multi-task learning. The authors think that “However, these Pareto optimization based MTL methods have not been fully investigated in the context of dynamic streaming tasks and are therefore unsuitable for CL tasks.”  The reviewer does not agree with this viewpoint. If we view the memory buffer data as multi-task data where each task takes some proportion, then the pareto optimal optimization in multi-task learning can be directly applied to continual learning.  The proposed method needs to compare with those state-of-art multi-task learning methods with multi-objective optimization, e.g. [2]. The novelty of the proposed method is not high from this viewpoint.


* Applying multi-objective optimization to continual learning is not new, as in [1], which already explores this research direction although the application field is different.







Reference:

[1] Multi-Objective Learning to Overcome Catastrophic Forgetting in Time-series Applications. 2022

[2] A Multi-objective Multi-task Learning Framework Induced by Pareto Stationarity. ICML 2022

**Questions:**

N/A

---

### Official Review · Reviewer_UYXS · 2023-10-23

**Soundness:** 2 fair
**Presentation:** 3 good
**Contribution:** 2 fair
**Rating:** 5
**Confidence:** 4

**Summary:**

This paper incorporates Pareto optimization into traditional gradient-alignment continual learning. Based on the reformulated gradient weighting problem, the authors proposed a novel Pareto regularized gradient alignment algorithm (PRGA), which considers the mutual influence among previous tasks to improve performance. PRGA achieves state-of-the-art across multiple datasets and different settings.

**Strengths:**

1. The paper is marginally well-written and easy to follow.
2. Intuitively, the idea of considering gradient interference among previous tasks would help improve the overall performance.
3. The proposed method achieves SOTA on several benchmark datasets.

**Weaknesses:**

My major concern is w.r.t the technical part, where the current gradient update for Pareto optimality seems inaccurate.
  * The solution of eq.(8) --- $g(\tilde{\lambda}^*)$,  is the weighted gradient that has the maximal minimum inner product with the gradients of previous tasks, indicating a minimal worst-case interference with these tasks.
  * However, $g(\tilde{\lambda}^*)$ is not related to $g_n$. Current update $u^*=g_n + g(\tilde{\lambda}^*)$ is not the solution to eq.(2) or eq.(3) without considering $g_n \cdot g(\tilde{\lambda}^*) > 0$ or $g_n\cdot g(\tilde{\lambda}^*) <0$.
  * To consider a similar objective, it needs to optimize $\max_u\min_{\alpha \geq 0} -\frac{1}{2} ||g_n -u||_2^2 + \alpha \langle u, g(\tilde{\lambda}^*)\rangle$. This gives $u^*=g_n + \alpha^* g(\tilde{\lambda}^*)$, where $\alpha^* = \arg\min\_{\alpha} \frac{1}{2}||g_n + \alpha g(\tilde{\lambda}^*)||_2^2$.  The optimal solution is as your discussion of RGA in Figure 3.

**Questions:**

Please see the previous section.

---

### Official Review · Reviewer_F5TA · 2023-10-30

**Soundness:** 3 good
**Presentation:** 3 good
**Contribution:** 2 fair
**Rating:** 5
**Confidence:** 5

**Summary:**

The paper addresses the challenge of catastrophic forgetting in continual learning (CL). It proposes a novel approach that leverages gradient alignment and Pareto optimization to enhance the performance of past tasks while ensuring the performance of the current task. The method, named Pareto Regularized Gradient Alignment (PRGA), is shown to outperform existing state-of-the-art continual learning methods across multiple datasets.

**Strengths:**

The paper introduces a fresh perspective by reformulating the gradient alignment optimization problem as a gradient weighting problem. The incorporation of Pareto optimality to capture the interrelationship among previously learned tasks is innovative. Comprehensive empirical results demonstrate the effectiveness of PRGA, showcasing its superiority over other methods.

**Weaknesses:**

The paper seems to lack a thorough discussion on the practical applications and limitations of Pareto optimization in continual learning. Comparisons with other gradient alignment-based methods are primarily performance-centric, without a deep dive into the conceptual differences and nuances of each approach.

**Questions:**

1)	The paper could benefit from a more detailed comparison with other gradient alignment-based methods, highlighting the unique advantages of PRGA.
2)	I can not understand the significance and interpretation of the confusion matrix presented in Figure 2 of the paper. While confusion matrices are typically used to visualize the performance of an algorithm on a classification task, the specific values and their implications in the context of this paper are not clear to me.
3)	I noticed that when employing the Pareto optimization approach, the focus seems to be primarily on the previously learned tasks without giving much consideration to the new task at hand. I'm curious as to why the new task isn't incorporated into the Pareto optimization process.
4)	In the paper, I noticed that certain comparative algorithms, specifically er and er-ace, were employed for benchmarking. However, I'm under the impression that there might be more recent state-of-the-art methods in this domain.
5)	I'm curious about the impact of the number of old tasks on the training of the continual learning model. Does an increase in the quantity of previously learned tasks influence the model's performance or its ability to learn new tasks effectively?
6)	I noticed that the paper does not seem to provide any accompanying code or implementation details for the proposed method. Could the authors consider making the code available for public access?

---

### Official Review · Reviewer_hYp1 · 2023-10-30

**Soundness:** 1 poor
**Presentation:** 1 poor
**Contribution:** 2 fair
**Rating:** 3
**Confidence:** 4

**Summary:**

This work studies the problem of catastrophic forgetting in continual learning from a gradient-alignment (GA) perspective. In said framework, one tries to alleviate catastrophic forgetting by following a direction similar to the current gradient, but such that its cosine similarity wrt past tasks remain positive. This work first reformulates GA and reinterprets it as a gradient reweighting (GR) problem, whose solution leads to their proposed regularized GA (RGA) algorithm. Then, the authors propose to solve a different GR problem which "takes into account the intrinsic correlations among tasks," leading to their proposed Pareto RGA (PRGA) algorithm. Finally, the authors test the proposed algorithms in different continual learning experiments, and compare them with an extensive number of baselines, showing that their algorithms lead to less catastrophic forgetting while obtaining good task performances.

**Strengths:**

- The problem of catastrophic forgetting is an important one and has the interest of the continual learning (CL) community.
- Reformulating gradient alignment as a gradient reweighting can be interesting for the CL community.
- The paper is well written, easy to follow for the most part, and most derivations and design decisions are explained in the manuscript.
- The experiments include numerous baselines, as well as statistical results beyond the average value.
- Empirical results are quite favorable for the proposed algorithm, which can be of interest for the practitioner.

**Weaknesses:**

I have several major problems with the current state of the manuscript. Since the scores in my review can look quite harsh, I will try my best to explain why I opted for these scores below:

**Soundness**
- W1. My score is based on the several claims and comments regarding Pareto optimality and multi-objective optimization (MOO). They simply make no sense (e.g. "as a crucial manner to achieve MOO", "we introduce the Pareto optimality to..."), and some claims are plainly wrong (e.g. "[Pareto optimality] seeks to maximize the overall marginal benefit across different tasks") and make me suspect that the authors did not fully grasp the meaning of Pareto optimality (e.g. any point in the Pareto frontier is Pareto optimal, and their losses can be, in general, as (im-)balanced as desired).

- W2. On a similar note, the term "hypergradient" seems to be used quite freely here, and I haven't been able to find a reference where hypergradients are defined a way similar to how it is done in Eq. 9. The two meta-learning citations use a different gradient to the best of my understanding.

**Presentation** While the paper is generally well-written, there are several typos, and citations and equations are not properly formatted. However, my low score relates with the "contextualization relative to prior work" and the disconnection between the proposed methods. I will describe now my biggest concerns:

- W3. Methods are not properly cited. For example,
  - MGDA is attributed to (Sener & Koltum, 2018) instead of the original author [1]. Sener & Koltum proposed MGDA-UB, an "upper-bound" of MGDA.
  - Again, (Sener & Koltum, 2018) is referenced together with another paper to cite the Frank-Wolfe algorithm, instead of citing the paper by Frank & Wolfe [2].
- W4. It is rather unclear how the two proposed methods are related (beyond solving two GR problems), and I feel the presentation sort of "forgets" about RGA at times. Indeed, RGA completely disappears from tables 3 and 5 and their corresponding discussions.

**Contributions** My bigger concerns here are three important ones:
- W5. **GR reformulation** I fail to see reformulating the problem as gradient re-weighting as a huge contribution as, to my understanding, the original derivations from (Lopez-Paz & Ranzato, 2017, Eq. 11) already define the problem as a GR problem.
- W6. **RGA** (minor) While the derivations are not super complicated, I find quite funny that all the derivations from section 3.1 are an almost one-to-one with those derivations from [3, App. A.3] (that solves a slightly-different problem)
- W7. **PRGA** This one I find it super concerning. PRGA (understood as the derivations from section 3.2) is far from novel, as it is _exactly_ the MGDA algorithm, popularized by Désidéri in 2012 [1], brought to the MTL community by Sener & Koltum in 2018, and first discussed (to the best of my knowledge) by Fliege & Svaiter in 2002 [4] (indeed, Eq. 5 is equivalent to Eq. 2 in their paper).

**Experiments** I have mostly focused on the theoretical part of this work, so I hope that my peers can complement my review in that regard. However, I have two concerns:
- W8. Given my discussion above, the effectiveness of the proposed methods seems to be a well-executed combination of existing techniques (which is valuable!). As such, it would be important to understand the degree of importance of each of these components, but I am afraid that the "iterative" ablation study from table 4 does not provide the necessary full-picture.
- W9. Again, I have not thoroughly looked into the different hyperparameters, but a quick look into [papers with code](https://paperswithcode.com/sota/continual-learning-on-cifar100-20-tasks) shows that the results for CIFAR100 with 20 tasks in CL has quite higher average accuracy (being the worst there 67%, and the best one here 33% in Table 1). As such, it would be necessary to clarify why such a difference in results, and whether conclusions hold despite the substantially lower performance.

---

[1]: Désidéri, J. A. (2012). Multiple-gradient descent algorithm (MGDA) for multiobjective optimization. Comptes Rendus Mathematique, 350(5-6), 313-318.

[2]: Frank, M. and Wolfe, P. An algorithm for quadratic programming. Naval research logistics quarterly, 3(1-2): 95–110, 1956.

[3] Liu, B., Liu, X., Jin, X., Stone, P., & Liu, Q. (2021). Conflict-averse gradient descent for multi-task learning. Advances in Neural Information Processing Systems, 34, 18878-18890.

[4] Fliege, J., & Svaiter, B. F. (2000). Steepest descent methods for multicriteria optimization. Mathematical methods of operations research, 51, 479-494.

**Questions:**

I don't have specific questions, besides everything I have discussed above.